# Barriers to Implementing Clinical Pharmacogenetics Testing in Sub-Saharan Africa. A Critical Review

**DOI:** 10.3390/pharmaceutics12090809

**Published:** 2020-08-26

**Authors:** Emiliene B. Tata, Melvin A. Ambele, Michael S. Pepper

**Affiliations:** 1Institute for Cellular and Molecular Medicine, Department of Immunology, and South African Medical Research Council Extramural Unit for Stem Cell Research & Therapy, Faculty of Health Sciences, University of Pretoria, Pretoria 0084, South Africa; u19109912@tuks.co.za (E.B.T.); melvin.ambele@up.ac.za (M.A.A.); 2Department of Oral Pathology and Oral Biology, Faculty of Health Sciences, School of Dentistry, University of Pretoria, PO BOX 1266, Pretoria 0001, South Africa

**Keywords:** clinical pharmacogenetics, pharmacogenetic testing, adverse drug reactions, genotype, phenotype, pharmacogene, barriers to pharmacogenetics implementation, Sub-Saharan Africa

## Abstract

Clinical research in high-income countries is increasingly demonstrating the cost- effectiveness of clinical pharmacogenetic (PGx) testing in reducing the incidence of adverse drug reactions and improving overall patient care. Medications are prescribed based on an individual’s genotype (pharmacogenes), which underlies a specific phenotypic drug response. The advent of cost-effective high-throughput genotyping techniques coupled with the existence of Clinical Pharmacogenetics Implementation Consortium (CPIC) dosing guidelines for pharmacogenetic “actionable variants” have increased the clinical applicability of PGx testing. The implementation of clinical PGx testing in sub-Saharan African (SSA) countries can significantly improve health care delivery, considering the high incidence of communicable diseases, the increasing incidence of non-communicable diseases, and the high degree of genetic diversity in these populations. However, the implementation of PGx testing has been sluggish in SSA, prompting this review, the aim of which is to document the existing barriers. These include under-resourced clinical care logistics, a paucity of pharmacogenetics clinical trials, scientific and technical barriers to genotyping pharmacogene variants, and socio-cultural as well as ethical issues regarding health-care stakeholders, among other barriers. Investing in large-scale SSA PGx research and governance, establishing biobanks/bio-databases coupled with clinical electronic health systems, and encouraging the uptake of PGx knowledge by health-care stakeholders, will ensure the successful implementation of pharmacogenetically guided treatment in SSA.

## 1. Introduction

Pharmacogenomics is an emergent but highly actionable form of personalised genetic medicine. Pharmacogenomics studies the impact of germline and somatic genetic variations (genotype) on drug response and the incidence of adverse drug reaction (ADR) phenotypes in an individual [1]. Clinical research has demonstrated the cost-effectiveness of pharmacogenetic (PGx) testing in improving drug compliance in patients, leading to decreased hospital admissions due to ADRs, especially for psychiatric patients on anti-depressants and anti-psychotics and cardiac patients on anti-platelet medication [1,2]. Furthermore, major PGx expert organisations such as the Clinical Pharmacogenetics Implementation Consortium (CPIC) [3] and the Dutch Pharmacogenetics Working Group (DPWG) [4] provide guidelines for PGx clinical implementation of gene–drug categories, so-called ”actionable variants” (gene variants with PharmGKB 1A or 1B high level of evidence) [5], with over 65 dosage recommendations in place. In addition, other expert organisations such as the EU-PIC (European Pharmacogenetics Implementation Consortium) [6], U-PGx (Ubiquitous pharmacogenomics) [7], RELIVAF (Latin American Network for Implementation and Validation of pharmacogenomics guidelines) [8], and SEAPharm (Southeast Asian Pharmacogenomics Research Network) [9] also provide pharmacogenetically-guided dosage recommendations. Indeed, the Food and Drug Administration (FDA) has published a list of PGx biomarkers for drug labelling with pharmacogenetically guided dosing [10].

The sub-Saharan African (SSA) region accounts for 25% of the global disease burden [11], with an increasing prevalence of non-communicable diseases (NCDs) [12] and emergent infectious diseases. Distinct and complex disease patterns amongst populations in the SSA region has led to distinct ADR patterns relative to Western and Asian countries [11]. Therefore, it becomes challenging for clinicians in this region with limited knowledge on potential drug–drug, drug–gene, and drug–drug–gene interactions when prescribing multiple medications to patients. The data on ADR incidence and the efficacy of most medications in populations of African descent are relatively scarce due to inefficient or absent pharmacovigilance programs [13]. There is however increasing evidence that the integration of PGx knowledge with other clinical data that influence drug response such as gender, age, weight, co-morbidities and lifestyle, will greatly assist clinicians in prescribing safe and efficient drug regimens to patients in SSA [14].

PGx Genome-Wide Association Studies (GWAS) have uncovered several population-based genetic variants (alleles) associated with ADRs. A majority of the variants (minor allele frequencies > 0.05) are recorded in most global populations [15]. Nevertheless, a few variants are rare (minor allele frequencies < 0.005) with varying global inter-ethnic frequencies that result in unique phenotypes in some populations [16]. Global inter-ethnic variability in genetic variants and drug response will mean that selected gene–drug pairs for clinical PGx testing in one population may not be very useful in another. For instance, testing of the loss of function *CYP2D6*17* allele, associated with amitriptyline-induced adverse effects, may serve as a useful marker for African and Latin American populations relative to European populations [17,18]. However, the genomes of African populations are greatly underrepresented in global GWAS studies [19]. Unique population growth, migration and genetic drift in the SSA region has resulted in high human genetic diversity and markedly lower but diverse linkage disequilibrium patterns between genetic variants across the region [20]. Therefore, risk scores of various genetic variants in SSA populations should not be inferred from European or Asian datasets, given the peculiarity of the genomic architecture in the African populations. Varying frequencies of genetic variants across different sub-populations in SSA might suggest inter-ethnic and inter-individual variability in drug response in this region [21]. A meta-analysis of GWAS on African cohorts have revealed novel *CYP2C9* and *VKORCI* gene variants with high genome-wide association, particularly in warfarin drug response, leading to subsequent dose adjustments for these cohorts [14]. This highlights the benefit and need to identify more African PGx markers through large-scale PGx research for PGx testing in SSA.

Genomic initiatives such as the African Pharmacogenetics Consortium (APC) [22] and H3Africa [23] have been created to harmonise PGx data and to create awareness of PGx research/testing in Africa. This has led to clinical and non-clinical PGx research on African populations that has characterised some unique PGx biomarkers, thereby demonstrating the potential benefits of integrating PGx testing in clinical practice in SSA [24,25,26,27,28,29]. A classic example is the characterisation of highly prevalent *CYP2B6*6* genetic variants in African populations associated with central nervous system toxicities in HIV patients on efavirenz treatment, which has led to specific drug dosage recommendations for African cohorts, relative to European populations [30].

Priority large-scale PGx clinical research and testing in SSA should involve “actionable variants” [15] associated with drug response in diseases contributing to the greatest morbidity and mortality such as tuberculosis, HIV, malaria, filaria, cancer, diabetes, cardiovascular diseases, and mental disorders [31,32]. Notably, priority research should be on the cytochrome P450 (CYP) family, including the *CYP1*, *CYP2*, and *CYP3* sub-families of genes, which encode proteins that are involved in the metabolism of approximately 90% of commonly prescribed medications [5]. The advent of cost-effective commercial genotyping microarrays with targeted pharmacogene panels such as the Axiom Precision Medicine Diversity Research Array (Thermo Fisher Scientific, Massachusetts, USA) and other FDA-approved arrays including the Gentris Rapid Genotyping Assay—*CYP2C9* and *VKORCI* (ParagonDx, LLC)—allows for the rapid testing of thousands of pharmacogenetically relevant variants. These arrays can be customised to include unique variants of African origin together with simplified bioinformatics workflows.

PGx testing has been successfully implemented in European and North American countries, mainly through large-scale initiatives, albeit with some limitations such as the complexities in accurately genotyping pharmacogenes and lethargy by test providers [33]. However, the clinical implementation of PGx testing in SSA primary health-care settings has been slow, highlighting the need for a review of some of the challenges involved. Factors such as under-resourced clinical health-care systems, limited PGx studies, scientific and technical barriers to genotyping pharmacogene variants, and socio-cultural and ethical issues regarding patients, clinicians, and health-care stakeholders have all been identified as potential barriers to the implementation of PGx testing in SSA. This review will comprehensively address these challenges with a focus on the scientific and technical barriers, and it will propose solutions that could potentially facilitate the clinical implementation of PGx testing in SSA.

## 2. Under-Resourced Clinical Health-Care Systems

The implementation of clinical PGx testing in SSA will assist physicians in tailoring drug regimens and dosages [1]. A case in point is the robust evidence indicating that testing for variants in pharmacogenes (*CYP2C9/VKORC1*) affecting warfarin response significantly reduces the incidence of ADRs [2]. Randomised clinical trials (RCTs) on PGx biomarkers in a population are the gold standard for obtaining robust evidence on the clinical effectiveness of PGx tests [1]. A clinical PGx test report typically comprises the individual’s genotype, predicted phenotype, and gene-guided dosing guidelines such as the CPIC guidelines. Pre-emptive testing involves genotyping an individual’s pharmacogenes before a drug is prescribed. The genotypes (usually multigene and multivariant panels) and extrapolated phenotypes are stored in a clinical Electronic Health Record (EHR) coupled to a point-of-care Clinical Decision Support System (CDSS) where a physician can access the results and subsequently implement pharmacogenetically guided regimens and dosages [1]. For reactive tests, a drug is first prescribed, and where necessary, this is followed by genotyping the individual, following which drug regimen adjustments are made. However, reactive tests tend to be more expensive, have low turnaround times, and fewer gene–drug pairs are included.

Clinical EHRs typically include patients’ demographic data, prescribed laboratory tests, prescribed medications, co-morbidities, and lifestyle data. Therefore, EHRs are critical in order to obtain longitudinal phenotype/genotype patient data for effective patient management, in addition to retrieving data for pre-emptive PGx testing and population-based studies. Furthermore, CDSSs, which typically contain recommended standardised PGx variant panels and automated dosage recommendations, are a prerequisite for effective pre-emptive PGx clinical testing [34]. However, the implementation of EHRs in the SSA clinical setting has been slow due to high cost, limited informatics infrastructure, lack of access and unreliable electricity supply, poor internet connection, and lethargy in the implementation of EHR by health-care stakeholders [34]. One study revealed that only 15 African countries have EHRs implemented in a few clinics, as most clinics still depend on paper-based patient health records [34].

Aquilante et al. recently demonstrated a hybrid model that facilitated the implementation of pre-emptive clinical PGx testing via the University of Colorado research biobank, coupled with efficient clinical EHRs and CDSS. This presents a unique opportunity for retrieving patient longitudinal genotype/phenotype data for dedicated PGx testing and research studies [35]. In this model, patients or healthy volunteers report to a clinic where informed consent is obtained, and a blood sample is collected. These are sent to the biobank, clinic, or research laboratory for DNA genotyping using a commercial Massarray, followed by a bioinformatic analysis. Structured PGx results are sent to a clinical EHR with patient phenotypes. Clinicians can subsequently access pharmacogenetically guided dosage recommendations based on CPIC guidelines from PGx-based CDSS tools [35]. We believe that this model could be successfully applied in in African countries with existing genetics research institutes. The feasibility of the proposed hybrid model will be made possible by engaging key stakeholders. Patients or healthy volunteers will need to be consented for DNA sample collection by clinicians and counsellors or trained community health workers. Then, clinician–geneticists will scan for evidence of PGx “actionable variants” from published literature and PGx expert guidelines to propose priority pre-emptive gene testing panels. The involvement of bioinformatics and information technology experts will be crucial in the design and setting up of a robust EHR linked to CDDS tools for easy access of PGx results by clinicians. Finally, hospitals and government leadership will need to ensure funding and logistical support, and promote the education of clinicians regarding implementing PGx testing workflows [35].

Biobanking activities are not well developed in Africa, which leads to the misrepresentation of African genetic data in global studies and databases. A biobanking and pharmacogenetics databasing initiative by African researchers at the African Institute of Biomedical Science and Technology (AiBST, Harare, Zimbabwe), catalogued 1488 DNA samples from inter-ethnic African populations, together with recorded frequencies of pharmacogene polymorphisms. Although donor clinical phenotypes were not recorded, validated clinically relevant genotype–phenotype associations could be extrapolated from the data [18]. Notably, the high frequency (14–34%) of the non-functional *CYP2D6*17* alleles recorded could have clinical relevance in anti-depressant and anti-psychotic therapy in African populations.

Recent research efforts have given rise to the establishment of more biobanks in some SSA countries such as the pan-African biobank (54gene) that has been set up in Nigeria to collect African genomic data and enforce the electronic capture of clinical data [36]. Other biobank initiatives including H3Africa, B3Africa, and the Global Emerging Pathogens Treatment Consortium [36] have curated African biospecimens and data in addition to enforcing biobanking policies in Africa. Hopefully, through these initiatives, the application of the hybrid method proposed will ensure the successful implementation of PGx testing in SSA in the near future.

## 3. Paucity of Clinical Pharmacogenetics Studies in SSA

Most RCTs demonstrating the clinical validity of PGx testing have been conducted on populations of European, American, African-American, and Asian descent [1]. There is a scarcity of PGx GWAS on the impact of rare genetic variants on drug response in African populations, with a few studies focusing on single gene–drug interactions. Indeed, data for pharmacogene variants for African populations are sometimes inferred from African-American populations; however, these populations have distinct ancestries and admixtures [20].

Global GWAS have commonly been employed to scan for multiple genetic variants in the human genome that are associated with drug response. Large effect sizes have been recorded for most variants linked to pharmacogenetic traits compared to other human traits. Notably large effect sizes have been established particularly for variants associated with warfarin, clopidogrel, and simvastatin therapy [16]. Candidate gene approaches have been successfully used to identify gene variants linked to drug response traits, although subsequent studies have failed to replicate previous results [37]. Therefore, drug response in humans is a complex and mostly a polygenic trait. Polygenic Risk Scores (PRS) have been developed to capture the effects of multiple combined variants across the human genome on disease risk and drug response [38]. Although the applicability of PRSs in clinical PGx testing is limited, PRSs have been applied clinically in evaluating the risk of developing diseases such as coronary artery disease and type-2 diabetes [39]. Nevertheless, few studies have demonstrated robust evidence for the utility of PRSs in statin and antidepressant drug response [40]. Importantly, the clinical utility of PRSs in PGx in African populations has not been evaluated given the scarcity of meta-analysis on African PGx data [38]. Therefore, PRS established on European and American populations have limited transferability to African populations given the disparity in population genetic architectures.

Several clinical studies on African populations (Table 1) have demonstrated the potential for PGx testing and dose adjustments in HIV patients receiving Efavirenz [26,27,28,30,41,42]. Other clinical studies have identified PGx biomarkers for African patients receiving rosuvastatin [43], imatinib [44], anti-retroviral (ARV) therapy, TB, and antimalarial comedication [25,45,46,47]. Importantly, a clinical study has demonstrated associations between genetic variants and clinical responses among Hepatitis C virus-infected patients from SSA and Europe treated with pegylated interferon-alpha/ribavirin [48]. Inter-ethnic variation in some PGx biomarkers, particularly variants in the cytochrome P450 genes, have been recorded in the SSA region. For instance, varying frequencies of some clinically relevant *CYP2B6*6* alleles in efavirenz drug response have been reported in Ugandan and Zimbabwean communities (68%) relative to South African populations (9%) [21]. Inter-ethnic variant variability could be attributable to environmental factors and differences in research designs giving rise to distinct patient cohorts. Therefore, distinctive inter-ethnic genotype–phenotype concordance should be taken into consideration in the development and clinical implementation of PGx testing.

A closer inspection reveals that most PGx research on SSA populations involves single gene–drug relationships; however, multigene–drug or gene panel testing provides a higher predictability of individual drug response. In addition, most studies have excluded pediatric populations, which is probably due to the existence of specific pediatric pharmacovigilance tools, including age-dependent drug dosage, relying on child-reported ADRs and the engagement of children and child care-givers in the research process [49]. The absence of clinical studies evaluating the economic value of implementing PGx testing in SSA health-care systems is also noteworthy, given that government and private insurers require evidence of the cost–utility of clinical tests for reimbursements. Therefore, there is an urgent need for dedicated PGx RCTs on populations of African ancestry to validate the impact of rare genetic variants on drug efficacy and ADRs while evaluating the cost-effectiveness of implementing clinical PGx testing. Although RCTs are the gold standard for providing robust evidence for clinical PGx testing, their cost-prohibitive nature calls for alternative approaches such as retrospective and prospective observational clinical studies.

The functionality of novel and rare variants uncovered in most of these studies (Table 1) have been predicted by computational algorithms. These algorithms utilise machine learning techniques based on a set of conserved variants associated with a disease; therefore, appropriate algorithms for PGx analysis need to be developed and trained on PGx variants [50]. This constitutes a challenge for the validation and subsequent inclusion of novel and rare variants in clinical testing. Experimental assays to validate the function of novel PGx variants remain the gold standard, although they are cost-prohibitive and time consuming [51]. Most experimental assays utilise knockout animal models such as mouse, Zebrafish, *Caenorhabdis elegans*, and *Drosophila*; however, the use of animal models is limited by cost together with the inaccurate extrapolation of human-specific drug disposition mechanisms. Complementary functional assays utilise transformed human cell lines, human-induced pluripotent stem cells, and organoids. For instance, stem cell-derived cardiomyocytes have been employed in identifying gene variants associated with doxorubin-induced cardiotoxicity in cancer patients. Notable advantages of employing human stem cells include the presence of human genomes that can easily be edited with tools such as CRISPR–Cas9 and the ability of these cells to differentiate into different tissues and organoids in culture [51]. The provision of more research funding and collaboration between interdisciplinary African researchers will boost robust PGx clinical research.

**Table 1 pharmaceutics-12-00809-t001:** Clinical studies aimed at validating pharmacogenetic biomarkers on clinical outcomes on sub-Saharan African populations.

Drugs	Clinical Study Outcome	References
**Efavirenz**	Pharmacogenetic determinants of response to Efavirenz in Black South African HIV/AIDS patients.	[41]
	Gender, weight, and *CYP2B6* genotype influence Efavirenz HIV/AIDS and TB treatment in Zimbabwe.	[26]
	*CYP2B6* variants impact plasma Efavirenz concentrations in HIV/TB patients in Tanzania.	[27]
	*CYP2B6* variants correlate with Efavirenz plasma concentrations in HIV patients in Zimbabwe.	[42]
	*CYP2B6* variants and pregnancy impact on Efavirenz plasma concentrations in Nigerian patients.	[28]
	Novel variants in pharmacogenes are associated with Efavirenz metabolism in HIV patients in South Africa.	[30]
	Composite *CYP2B6* alleles are significantly associated with Efavirenz-mediated central nervous system toxicity in HIV patients in Botswana.	[52]
**Nevirapine**	*CYP2B6* and *CYP1A2* variants impact Nevirapine plasma concentrations and HIV progression respectively in an HIV patient cohort in Zimbabwe.	[29]
**PEGylated Interferon-alpha/Ribavirin**	*IL28B* SNPs correlate with treatment response in Hepatitis C patients from SSA.	[48]
**ARV/TB**	GWAS study identified SNPs linked to drug-induced hepatoxicity in HIV/TB patients in Ethiopia.	[47]
**ARV/TB/Antimalarials**	*CYP2B6*6* variant and Efavirenz concentration impact on Lumefantrine plasma levels in HIV/Malaria patients in Tanzania.	[25]
	High frequency of the *CYP2B6*6* allele is associated with poor clinical response in HIV/TB/Malaria patient cohort in Congo.	[46]
**Lumefantrine**	*CYP3A4*, *CYP3A5* variants impact Lumefantrine response in a cohort of pregnant women with malaria in Tanzania.	[25]
**Imatinib**	*CYP3A5*3* and *ABCB1 C3435T* variants influence clinical outcomes and plasma concentrations of Imatinib in Nigerian patients with chronic myeloid leukaemia.	[44]
**Risperidone**	*CYP2D6* variants did not significantly impact the incidence of ADRs in a South African cohort.	[53]
**Amitriptyline**	*CYP2D6* variants influence ADR incidence in patients with painful diabetic peripheral neuropathy in a South African cohort.	[54]
**Rosuvastatin**	Specific pharmacogene variants influencing rosuvastatin response in African populations.	[24]
**Warfarin**	*CYP2C9* and *VK0RC1* variants are associated with dose–response in Warfarin-treated Sudanese patients.	[55]
	Novel *CYP2C9/VK0RC1* variants influence Warfarin response in a black South African cohort.	[56]
	*CYP2C9/VKORC1* variants did not correlate with Warfarin dose–response in a Ghanaian cohort.	[57]

Co-infection and disease co-morbidity patterns are not uniform across the SSA region, which constitutes a major challenge for disease management. Co-morbidities such as malaria, neglected tropical diseases, HIV, and TB are commonly observed in Central and West Africa, while in the southern region, co-morbidities such as HIV and TB are prevalent [11]. Therefore, treatment regimens and ADR patterns for one region cannot be extrapolated to another. Furthermore, the rising incidence of NCDs such as cancer, diabetes, cardiovascular disease, and neuropsychiatric disorders [12] has complicated the burden of comorbidities, leading to multidrug regimens being prescribed to patients. Data on the frequency of ADRs and the efficacy of medications for the treatment of co-morbidities on African populations are scarce [13]. ADRs may lead to patient non-compliance and prolonged hospital stays, thereby placing a cost burden on already strained health systems.

Pharmacovigilance systems aimed at monitoring drug safety are underdeveloped or even non-existent in some SSA countries. Therefore, clinicians are not aware of the exact framework for communicating ADRs to health institutes or national health departments. Insufficient funding and lack of communication between clinicians and national health departments, as well as limited physician competency on pharmacovigilance, greatly contribute to underreporting of ADRs in SSA countries [13] (Figure 1). Most clinicians complain of a lack of time to prepare ADR reports, while some physicians in private health care might be lethargic in reporting ADRs due to fear that submitting inadequate reports may lead to legal action taken either for medical malpractice or incompetence [58]. Under-reporting of ADRs is reflected in the under-participation by SSA countries in pharmacovigilance programs where only 35 countries actively participate in the International Drug Monitoring Program run by the by the World Health Organisation [13].

The few studies on the impact of ADRs on affected populations and health care systems in SSA have demonstrated that ARVs contribute to 80% of ADRs and that ADRs from some antibiotics are up to 10% higher in African populations relative to other populations globally [13]. A case in point is the high frequency of ADRs such as cardiotoxicity and congestive heart failure commonly observed during anthracycline treatment in Africans relative to Caucasian populations [59]. Clinical trials on medications used to treat NCDs have mainly been carried out on populations other than African, leading to rare ADRs being recorded in Africans. Several factors could be attributable to the unique patterns of ADRs in SSA populations including host genetic factors, age, weight, polypharmacy, lifestyle, and the utilisation of counterfeit or expired medication [13]. Furthermore, there is a reported low efficacy of drugs such as beta blockers and angiotensin-converting enzyme inhibitors in African populations [60]. Therefore, large-scale PGx GWAS in African populations are needed to uncover drug–gene, drug–drug, drug–drug–gene interactions, as well as gene loci impacting ADRs and the low efficacy of some drugs.

## 4. Challenges in Genotyping Pharmacogene Variants

Drug response phenotypes are primarily determined by mechanisms involved in the induction or inhibition of enzymes, as well as the functionality of transporters and other proteins involved in absorption, distribution, metabolism and excretion (ADME) (pharmacokinetics). The pharmacodynamics of interactions of a drug with its target or other molecules in disease pathways also impacts drug response. Together, polymorphisms mostly in the coding and regulatory regions of genes for these enzymes account for approximately 25% of the variability in inter-individual and inter-ethnic drug responses [61]. Other factors affecting the activities of these enzymes include epigenetic regulation, gender, age, lifestyle, and concomitant medications [62].

The Cytochrome P450 family of enzymes (including CYP1A2, CYP2D6, CYP2C9, CYP2B6, CYP2C19, CYP3A4, and CYP3A5) coded for by their respective pharmacogenes is responsible for the metabolism of almost 90% of prescribed medications [3]. Other important pharmacogenes with “actionable variants” include *SLCO1B1*, *VKORC1*, *DPYD*, *TPMT*, *NUDT15*, *HLA-A*, and *HLA-B*. Together, the “actionable variants” of pharmacogenes listed above impact the metabolism of up to 49 commonly prescribed drugs used in primary care in SSA and globally (including anti-infectives, antihypertensives, antilipidemic, antidepressants, and anticancer). The CPIC and DPGWG assigns “actionable variants” based on sufficient clinical evidence while providing gene–drug dosing guidelines [15] (Figure 2).

Germline variants in genes coding for these enzymes (Figure 2) can be SNPs (single nucleotide polymorphisms), INDELS (insertions and deletions), and copy number variations (CNVs) including duplications, deletions, and complex structural variants (SVs) [5]. These variants result in alleles that confer different phenotypes. Specifically, phenotypes resulting from variants in the CYP2 family of enzymes are grouped into four main categories, namely (a) ultra-rapid metabolisers (UM)—carry two or more gain-of-function alleles including gene duplications; (b) normal metabolisers (NM)—carry normal-function gene alleles; (c) intermediate metabolisers (IM)—carry one non-functional allele; and (d) poor metabolisers (PM)—carry two non-functional alleles, including gene deletions [5]. PMs normally experience more ADRs in the case of the metabolism of an active drug due to high plasma levels of the active compound, while UMs will experience therapeutic failure due to the rapid metabolism and clearance of the active compound from their systems. Globally, with respect to the CYP2D6 enzyme, 0.4–5.4% of individuals are PMs, 0.4–11% are IMs, 67–90% are NMs, and 1–21% are UMs [61]. For example, the *CYP2D6**2XN allele found in UM individuals is recorded in 1–16% of Africans, while *CYP2D6*17* found in PM individuals is recorded in 35% of African populations [21]. The high frequency of recorded PM individuals in African populations is of great clinical significance, as most of the commonly prescribed medications and food substances are metabolised by the CYP2D6 enzyme.

Global inter-ethnic variability in the frequency of PGx “actionable variants” is evident for some genes (Table 2), leading to the clustering of biogeographical populations into European, Asian, and African. The frequency of African-specific PGx biomarkers as highlighted (Table 2) reveals some population-specific variants: for instance, the high frequency of the *CYP2D6*17* allele in African populations relative to European and Asian populations. Nevertheless, the clinical relevance as well as frequencies of some “actionable variants” for specific medications have not been characterised in African populations, as is evident from the absence of frequency data on the CPIC database. Dedicated clinical PGx research with large sample sizes of African cohorts might hopefully uncover region-specific genotype–phenotype correlations.

In addition to germline variants, somatic variants have been characterised and catalogued in relation to cancer treatment responses. For instance, somatic mutations on *EGFR* and *BCR-ABL* genes are highly predictive of gefitinib and imatinib drug response respectively in non-small cell lung cancer patients of European ancestry [63]. Furthermore, few studies have demonstrated inter-ethnic variability in the frequency of some sensitising somatic variants. However, studies on somatic variant profiles in cancer patients in African populations are sparse [63]. Neoplasms are one of the leading causes of deaths from NCDs in the SSA region [12]. Therefore, it is crucial for health-care stakeholders to start prioritising PGx research and effective PGx clinical interventions in cancer patients.

Transient drug induction or the inhibition of ADME enzymes during co-medication is known as phenoconversion. The phenomenon of phenoconversion can also be a result of inflammatory processes in the body and can greatly impact the interpretation of PGx test results [64]. Very few clinical studies globally, and in Africa in particular, have demonstrated the impact of phenoconversion on PGx test result interpretation. An in vitro investigation on CYP2C19 enzyme activities in human donor liver microsomes revealed that the inclusion of phenocopying factors significantly improved phenotype prediction [65]. Phenoconversion is most likely to influence PGx results in African populations due to complex health interventions that include herbal medicines, although some enzymes such as CYP2D6 are not easily induced. Khat (*Catha edulis Forsk*), a psychoactive herb commonly used in East Africa, has been identified as a potent inhibitor of the CYP2D6 enzyme (which metabolises up to 25% of prescribed medications) [66]. Therefore, environmental factors such as regional lifestyles and co-medication should be considered in future studies and in the interpretation of clinical pharmacogenetically guided drug prescription.

The advent of improved techniques including next-generation sequencing (NGS), Sanger sequencing, and microarray genotyping techniques has completely revolutionised genotyping and GWAS, making genotyping more cost-effective, high-throughput, and accessible for clinical use [67]. Novel rare variants (including SNPs, CNVs, and complex hybrid SVs) in pharmacogenes such as the *CYP2D6* gene, with important functional effects, have recently been identified using NGS, highlighting the complexities in these genes [68]. These rare variants are thought to be differentiated amongst populations, which warrants more research on African rare variants, given the diverse genetic pool recorded in this region [20]. The high genetic diversity may result in varying efficacy and ADRs reported in African populations. South Africa in particular has a unique and complex genetic population, resulting from admixture between the native Khoisan with Bantu and European populations [20]. Recently, deep NGS of the pharmacogenes of a Bantu-speaking cohort in South Africa revealed rare novel variants with predicted functional effects that have not been recorded in other African populations [69]. This includes the identification of novel deleterious variants in the flavin-containing monooxygenase 2 gene, which is involved in the oxygenation of sulphur-containing drugs in humans [69]. Distinct and highly diverse alleles of the Cytochrome P450 family have been recorded in African populations relative to other populations, highlighting the need for further dedicated PGx and functional studies on these unique variants [21]. For instance, the loss-of-function *CYP2B6*6* allele which accounts for low efavirenz plasma levels and an increased risk of neurotoxicity, is highly prevalent in African populations relative to Europeans and Asians [30].

Novel and efficient genotyping platforms provide an opportunity for research initiatives to catalogue the structure and clinical functional effects of rare African genetic variants. Only a few private diagnostic and university research institutes in SSA are equipped with genotyping and bioinformatics technologies, while access to PGx tests is essentially limited to the private health-care sector. The cost-prohibitive nature of genetic testing and the general lack of expertise constitute barriers to setting up genetic testing laboratories. Furthermore, there are no standardised gene panels or guidelines for clinical testing between the few laboratories involved. Finally, there is lethargy in obtaining laboratory accreditation for genetic testing in countries that offer direct-to-customer testing. This stems from the absence of national guidelines for genetic testing in most SSA countries and other less developed regions [17].

Accurate genotyping and phenotype translation into actionable clinical decisions requires state-of-the-art sequencing technologies and computational platforms. Notably, the recent single-molecule real-time (SMRT) NGS platform has uncovered novel and complex SVs of the *CYP2D6* gene with predicted functional impact. Commercial genotyping arrays coupled with bioinformatics algorithms have been designed to incorporate millions of SNPs on chips, leading to high-turnaround times. SNPs (tag SNPs) incorporated in the array are selected such that they represent multiple other SNPs in the genome based on their linkage disequilibrium. Nonetheless, genomic data on most commercial arrays is based on data principally from Caucasian and Asian populations. Therefore, rare variants of African origin may not be captured using pre-designed arrays, due to differences in the linkage disequilibrium patterns between populations. This challenge is reflected in low specificities and sensitivities being recorded when these arrays are applied on African samples [67]. The challenge of genotyping novel rare variants can be overcome by using phasing and imputation software to extrapolate missing variants from whole-genome databases and subsequently customising the pre-designed array.

A H3Africa chip-based genotyping array with tag SNPs of clinically important pharmacogenes based on African genome sequences is now available for research [23]. This array will be a more accurate genotyping tool for African studies following its validation, as opposed to arrays that do not specifically include African variants. Importantly, different arrays have different sets of variants leading to the non-standardisation of variant panels tested [67]. Clinicians are faced with the challenge of selecting the most appropriate genotyping technology. There are several factors to be considered when selecting an appropriate genotyping test, including turnaround time, ability to detect/customise multiplexed variants across global ethnicities, and ease of workflow [67]. Varying sensitivities and specificities have been recorded by different genotyping platforms, leading to inconsistent variant calls. For example, inconsistent CNV and SV detection for *CYP2D6* has been recorded by different genotyping platforms, depending on their design [67,68]. Therefore, multiple genotyping platforms must be utilised for accurate phenotype prediction, which imposes a cost burden. The functional impacts of rare CNVs and SVs in *CYP2D6*, *CYP2C19*, *CYP2A6*, *SULTIA1,* and *GSTT1* have been identified with varying inter-ethnic frequencies (https://www.pharmgkb.org). These rare variants might explain the complex phenotypes unaccounted for by the “missing heritability” issue recorded in most GWAS studies. Therefore, uncovering the frequency and functional impact of pharmacogene CNVs and SVs on drug disposition in populations of African descent will greatly improve the accuracy of metaboliser status determination in clinical studies. Other challenges in genotyping include recorded allelic drop-out by different assays and the inability of some genotyping assays to determine which allele is duplicated, particularly with respect to the *CYP2D6* gene [68]. The accurate genotyping of complex PGx genes usually requires the utilisation of multiple genotyping techniques including those that are PCR-based, mass arrays, and sequencing techniques [68].

The absence of a consensus in the translation of genotyping results to actionable drug prescription presents another challenge for PGx test result interpretation. For example, the phenotypic effects of loss-of-function and gain-of-function *CYP2C19* alleles are drug-dependent [5]. Although the CPIC is continually updating guidelines for drug–gene interactions and translating “actionable variants” into phenotypes based on activity scores, clinical validity of these recommendations is still required, particularly in SSA populations. Furthermore, some guidelines for translating *CYP2D6* genotypes into actionable clinical decisions are divergent between expert organisations such as the CPIC and DPWG, although efforts are being made towards harmonising guidelines. The functional impacts of rare novel alleles are commonly predicted using computational algorithms, but experimental studies remain the gold standard. This further highlights the need for more in vivo studies on the functional impact of novel rare African gene variants. Experimental assays to validate the functionality of the novel variants could be performed by employing whole animal models, human transformed cell lines, and organoids, depending on available funding and logistics [51].

Recently, an assembled pan-African reference human genome from sequences of African individuals revealed an additional 296.6 Mb of unique sequences relative to the current reference human genome [70]. Thus, the current reference human genome is not appropriate for African PGx and genetic studies. Genomic initiatives in Africa such as H3Africa [23] and APC [22] have been supporting PGx research in Africa by sequencing and curating the genomes of African individuals, supporting genomic research capacity building and harmonising genotype and phenotype data recording. This has led to an increase in integrated capacities for PGx research as well as an increase in the utilisation of genomics and bioinformatics technologies in SSA. Other genomic initiatives such as the Southern African Human Genome Programme [71], the African Genome Variation Project [72], and the MalariaGEN project [73] have provided databases of African genome sequences. Although governments in SSA are compelled to channel their limited resources towards the fight against infectious diseases, the cataloguing of African PGx data will contribute to diagnostic and drug development pipelines tailored for African populations.

## 5. Socio-Cultural and Ethical Challenges vis-à-vis Clinicians

The successful implementation of PGx testing will require acceptance and adequate knowledge of PGx by health-care workers, especially physicians. Nevertheless, competency on PGx testing amongst clinicians in African populations is lacking, which has also been reported as one of the major barriers for implementing PGx in other under-resourced clinical settings such as in Latin America [17,74]. The lack of competency stems from the absence of or limited PGx training programs in health-care training institutions and universities in Africa. In addition, physicians are not aware of the available evidence and curated guidelines for PGx testing implementation [74,75]. The CPIC and PharmKGB databases are excellent resources for clinicians to acquire adequate guideline support for priority PGx testing implementation. PGx programs should be a prerequisite in health training schools and university curricula in SSA countries. The ordering, analysis, and interpretation of PGx test reports are complex, requiring access to and the incorporation of PGx data into EHRs and CDSS. Therefore, physicians require adequate knowledge concerning the logistics involved in ordering and interpreting PGx tests. Importantly, clinicians also lack confidence when counselling and/or recommending PGx tests to patients, reflecting lethargy in updating themselves regarding PGx analysis and research from peer-reviewed literature [74]. Furthermore, the absence of clear regulations for genetic testing coupled with cost-prohibitive tests in private and public laboratories in SSA greatly contributes to the non-ordering of these tests by physicians [74]. Most physicians in SSA clinics are not aware of the ethical and legal implications of returning PGx testing results to patients, stemming from the absence of national regulatory guidelines for genetic testing [74,76]. Genetic counselling of patients is required before returning PGx test results to patients. However, the absence of genetic counsellors in most public clinics in SSA places a high burden on physicians. Importantly, physicians may be wary of fatal outcomes, in the case where an inaccurate drug regimen or dosage was selected based on genotype [70]. The sharing of secondary or incidental findings of disease-related genes during PGx testing with an individual or family also poses an ethical issue. Indeed, patients need to be assured of the privacy and confidentiality of their results, especially with respect to employer and insurer decisions. Furthermore, the right to ownership of patient data and samples varies depending on the country’s policies, which needs to be known and acknowledged by clinicians. The unique consenting procedure for PGx testing in SSA populations is also noteworthy. It has been suggested that informed consent for African populations needs to be modelled relative to the culture and ethics of the communities and not extrapolated from Western cultures [74,77]. A tiered informed consent involving the use of African colloquialisms to explain hereditary has been suggested for use in African cohorts [77]. Overcoming these observed social and ethical barriers will require collaboration between clinicians, genetic councillors, and research experts to provide robust institutional support for the successful implementation of PGx testing.

## 6. Socio-Cultural and Ethical Challenges vis-à-vis Patients

Knowledge and awareness of PGx testing by patients and caregivers in SSA populations is absent. This might lead to an unacceptability of PGx testing, as most patients from rural areas with limited education and socio-economic status will lack understanding, including misconceptions about the costs and invasiveness of the tests. Most patients do not obtain additional information on the benefits or logistics of PGx testing, and therefore, they rely more on the physician to make final decisions for them [74]. Some patients might be reluctant to perform a PGx test for psychological reasons, based on the implications of the results. This might stem from religious and cultural beliefs regarding genetic material [74]. Most African people do not have specific words in their native languages describing genetic material. Furthermore, many consider genetic material to be related to paternity and ancestry, which may affect their understanding and acceptability of PGx testing. Therefore, physicians need to implement traditional and religious symbols to facilitate patient understanding of PGx tests. Patients from rural areas in SSA countries do not have access to PGx testing services due to their cost-prohibitive nature, and only a few private laboratories offer these services to private patients in urban areas [74]. Patients and caregivers will appreciate the benefits of PGx testing in their clinical care if the tests are implemented in most clinics and are affordable.

## 7. Socio-Cultural and Ethical Challenges vis-à-vis Health-Care Authorities and Insurers

The implementation of clinical PGx testing in SSA poses a financial burden on already challenged public health-care systems. Health-care policy makers and government departments need robust evidence that demonstrates the cost-effectiveness of PGx testing implementation, as resources are usually directed towards urgent public health-care issues. Therefore, most researchers and clinicians rely on foreign and private funding for genetic research and testing. This highlights the need for more large-scale clinical studies in SSA populations aimed at demonstrating the cost-effectiveness and ability of PGx tests to improve the quality of life of patients. Government health departments also exhibit lethargy in engaging researchers and providing informatics support to clinics, as PGx tests are perceived to be expensive [74].

Most countries, particularly in SSA and other less developed regions [17], lack specific and clear regulatory policies for the implementation of genetic testing, and in particular PGx testing. For instance, only South Africa, Nigeria, and Malawi amongst SSA countries provide clear and specific guidelines for genetic testing and research [78]. The Academy of Science of South Africa has provided a review of ethical, legal, and social implications of genetics research and testing in South Africa [79]. Nevertheless, a final framework on data sharing and genotyping test accreditation can only be provided by the national health and science research departments. This poses a challenge for researchers and funders involved in PGx research and implementation. The implementation of PGx testing regulations in SSA countries will depend on the continued training of geneticists, setting up of national genetic testing infrastructure, and research funding from government health departments.

The non-reimbursement of PGx tests in SSA countries poses another challenge for clinical implementation. Insurers, particularly those in the private health-care sector, require standard clinical guidelines for frequent use by physicians and evidence of cost–utility for their coverage of PGx tests [74]. Given the absence of clinical studies demonstrating the cost-effectiveness and clinical utility of PGx in African populations, most insurers in this region will not provide coverage for these tests. Therefore, cost–utility analyses of implementing PGx testing in SSA populations needs to be undertaken in order to demonstrate clinical utility and provide motivation for reimbursement by health-care insurers.

## 8. Conclusions and Future Directives

The clinical utility and cost-effectiveness of PGx testing for improved patient health care is increasingly being demonstrated. However, the implementation of PGx testing in SSA is still lagging. This review highlights several challenges that need to be surmounted for the future implementation of routine PGx testing in SSA. These include the establishment of robust clinical health-care systems, investing in dedicated PGx studies and governance, improving scientific and technical barriers to genotyping pharmacogene variants, and PGx knowledge uptake by health-care stakeholders.

We believe the implementation of pre-emptive clinical PGx testing in SSA countries is feasible through a hybrid model that incorporates patient genetic data from research biobanks linked to other clinical data in EHS/CDDS. This will involve the input of multiple key stakeholders, including patients, clinicians, geneticists, information technology specialists, and health departments. Furthermore, this model will also provide a unique opportunity for the easy retrieval of patient phenotypic and genetic data for large-scale GWAS PGx research initiatives. Proactive strategies such as the provision of institutional support by national health departments will ensure the strengthening of health-care systems in SSA countries.

Clinical PGx research in some SSA countries has uncovered rare variants in African populations with a significant functional impact. Preliminary data from clinical studies demonstrate the benefits of implementing PGx testing in SSA for optimal patient care. Additional robust large-scale studies on populations of African ancestry will provide strong evidence for the cost-effectiveness and clinical utility of PGx testing in these populations. Robust evidence on the cost-utility of PGx testing will ensure support for clinical PGx testing implementation from health-care departments, policy makers, and health-care insurers. Furthermore, the genotypes of large populations of Africans should be catalogued by employing a combination of cost-effective high-throughput NGS and customisable massarray genotyping techniques coupled with bioinformatic analysis. Finally, the provision of additional funding and regulations for PGx research and clinical diagnostic laboratories will ensure increased expertise and accessibility to genotyping techniques in SSA.

The availability of published CPIC and PharmKGB guidelines for pharmacogenetically guided prescription provides an excellent resource for PGx testing knowledge uptake by clinicians, counsellors, and subsequently patients and caregivers in SSA. Finally, we suggest that the continual curation of clinical PGx testing evidence, the setting up and harmonisation of regulatory policies, and the education of health-care stakeholders across SSA countries by African genomic initiatives such as the H3Africa and the African Pharmacogenetics Consortium (APC) will facilitate the implementation of PGx testing in SSA.

## Figures and Tables

**Figure 1 pharmaceutics-12-00809-f001:**
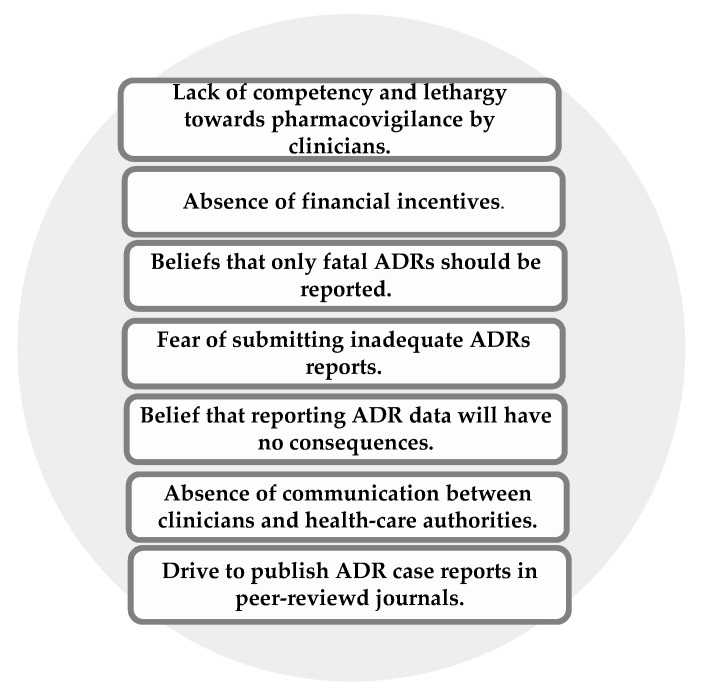
Main factors contributing to under-reporting of adverse drug reactions (ADRs) by clinicians in sub-Saharan African (SSA) countries.

**Figure 2 pharmaceutics-12-00809-f002:**
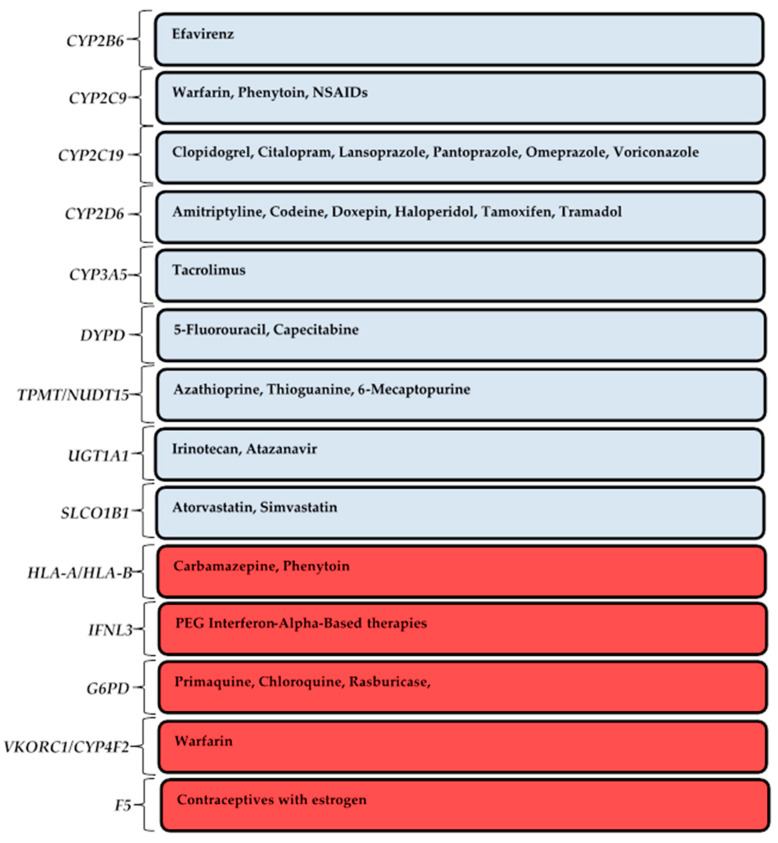
Pharmacokinetic (blue), pharmacodynamic (red) gene–drug pairs with Clinical Pharmacogenetics Implementation Consortium/Dutch Pharmacogenetics Working Group (CPIC/DPWG) “actionable variants” and dosage guidelines in commonly prescribed medications that will benefit patients in primary care in SSA.

**Table 2 pharmaceutics-12-00809-t002:** Average frequencies (%) of alleles (* star alleles) with “actionable variants” in pharmacogenes as assigned by CPIC in major global biogeographical populations. Average frequencies are based on the reported frequencies in one or multiple studies [15].

Gene	Allele	Functional Effect	Sub-Saharan Africa	African American/Afro-Caribbean	Caucasian	Central/South Asian
***CYP2B6***	***4**	Increased function	**0.0000**	0.0103	0.0409	0.0990
	***5**	Normal function	**0.0200**	0.0621	0.1155	0.0110
	***6**	Decreased function	**0.3749**	0.3170	0.2330	0.1850
	***9**	Decreased function	**-**	0.0465	0.0147	0.0590
	***16**	Decreased function	**0.0054**	0.0000	0.0000	0.0000
	***18**	No function	**0.0577**	0.0330	0.0000	0.0000
***CYP2C9***	***2**	Decreased function	**0.0131**	0.0224	0.1273	0.1138
	***3**	No function	**0.0112**	0.0301	0.0763	0.1099
	***5**	Decreased function	**0.0131**	0.0116	0.0003	0.0000
	***11**	Decreased function	**0.0257**	0.0139	0.0016	0.0010
***CYP2C19***	***2**	No function	**0.1568**	0.1815	0.1466	0.2699
	***3**	No function	**0.0027**	0.0028	0.0017	0.0157
	***4A/B**	No function	**0.0000**	0.0000	0.0020	0.0000
	***5**	No function	**0.0000**	0.0000	0.0000	0.0032
	***6**	No function	**0.0000**	0.0000	0.0003	0.0006
	***8**	No function	**0.0000**	0.0011	0.0034	0.0000
	***9**	Decreased function	**0.0270**	0.0143	0.0007	0.0001
	***10**	Decreased function	**0.0000**	0.0033	0.0000	0.0001
	***17**	Increased function	**0.1733**	0.2072	0.2164	0.1708
***CYP2D6***	**2XN**	Increased function	**0.0173**	0.0188	0.0084	0.095
	***3**	No function	**0.0015**	0.0032	0.0159	0.0011
	***4**	No function	**0.0338**	0.0482	0.1854	0.0906
	***5**	No function	**0.0338**	0.0482	0.1854	0.0459
	***6**	No function	**0.0000**	0.0029	0.0111	0.0000
	***8**	No function	**0.0000**	0.0000	0.0002	0.0000
	***9**	Decreased function	**0.0000**	0.0044	0.0276	0.0300
	***10**	Decreased function	**0.0557**	0.0382	0.0157	0.0867
	***14**	Decreased function	**-**	0.0000	0.0000	-
	***17**	Decreased function	**0.1929**	0.1688	0.0039	0.0007
	***41**	Decreased function	**0.1147**	0.0372	0.0924	0.1230
***CYP3A5***	***3**	No function	**0.2409**	0.3160	0.9249	0.6733
	***6**	No function	**0.1932**	0.1112	0.0015	0.0000
	***7**	No function	**0.0864**	0.1200	0.0000	-
***DPYD***	***2A**	No function	**0.0000**	0.0031	0.0079	0.0051
	***13**	No function	**0.0000**	0.0000	0.0006	0.0000
	**2846A > T**	Decreased function	**-**	0.0031	0.0037	0.0006
	**1236G > A**	Decreased function	**0.0000**	0.0031	0.0237	-
***TPMT***	***2**	No function	**0.0000**	0.0053	0.0021	0.0002
	***3A**	No function	**0.0016**	0.0080	0.0343	0.0042
	***3B**	No function	**0.0000**	0.0000	0.0027	0.0017
	***3C**	No function	**0.0529**	0.0240	0.0047	0.0112
***NUDT15***	***2***	No function	**-**	-	0.000	0.035
	***3**	No function	**-**	-	0.002	0.061
	***6**	Uncertain function	**-**	-	0.003	0.013
	***9**	No function	**-**	-	0.002	0.000
***SLCO1B1***	***5**	Decreased function	**0.0000**	0.0000	0.0083	0.0224
	***15**	Decreased function	**0.0297**	-	0.0439	0.1214
	***17**	Decreased function	**-**	0.1330	0.0519	-
***UGT1A1***	***28**	Decreased function	**0.4000**	0.3734	0.3165	0.4142
	***6**	Decreased function	**0.0000**	0.0040	0.0079	0.0449
	***37**	Decreased function	**0.0371**	0.0570	0.0007	0.0000
***HLA-A/HLA-B***	**HLA-A*31:01**	High risk allele	**0.52**	0.98	2.84	2.20
	**HLA-B*15:02**	High risk allele	**0.00**	0.10	0.04	4.64
	**HLA-B*57:01**	High risk allele	**0.79**	0.10	3.23	4.49
	**HLA-B*58:01**	High risk allele	**5.54**	3.89	1.32	4.54
***IFNL3***	**IL28B:CC**	Increased response	**26.8**	15.2	36.5	1.9
	**IL28B:CT**	Increased response	**52.4**	40.62	47.6	23
	**IL28B:TT**	Increased response	**20.8**	43.75	15.9	75.1
***G6PD***	**376A>G**	Deficiency	**0.312**	-	0.0595	-
***VKORC1***	**1639G>A**	Decreased Warfarin dose	**12.900**	10.274	41.2242	15.317

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
