# Peer review of "Barriers to Implementing Clinical Pharmacogenetics Testing in Sub-Saharan Africa. A Critical Review"

_pharmaceutics, 2020, doi:10.3390/pharmaceutics12090809_

Round 1

Reviewer 1 Report

Thank you for this remarkable effort and contribution to the field. I do appreciate the opportunity to review this manuscript and firmly believe this review paper, with consideration of the comments below, would be a valuable addition to the scientific literature.

  1. When we consider the number of individuals included in GWAS conducted so far, based on ethnicity, only ~2% are Africans. Furthermore, Africans seem to be disproportionately affected by poor health conditions (i.e., SSA region accounts for 25% of the global disease burden [page 2, Introduction], despite they represent a smaller fraction of the world’s population), but they remain underrepresented in pharmacogenomic studies. Since GWAS findings may not replicate across different ethnic groups, such a disparity is unacceptable. Please, discuss this major concern in the review.
  2. As correctly mentioned by the authors, the ability to replicate genetic associations across diverse populations can be markedly affected by several factors including differences in linkage disequilibrium (LD) across ethnicities (or parental/ continental populations) that may influence how well causal variants are captured by tagging SNPs identified in a single population. Africans have maintained larger and more sub-structured populations resulting in diverse patterns of LD across the continent. Because of such a peculiar genomic architecture in Africans, investigators may be able to narrow down the region that harbors causative alleles. Discuss this potential advantage to fine-map causative variants in Africans.
  3. Please, discuss the potential role of differential effect sizes (Δ) of actionable SNPs or even polygenic risk scores (PRS) across diverse ethnic groups (i.e., Europeans vs. Africans) on the ability to replicate pharmacogenetic associations across diverse populations (i.e., transferability) and how this limitation on transferability might underestimate or overestimate the risk of ADRs or poor responses to drugs.
  4. Introduction, page 2 of 17, second paragraph, last sentence: “This has led to clinical and non-clinical Pgx research on African populations that have characterized some unique Pgx biomarkers, and…” Please provide some examples of these “unique Pgx biomarkers” to illustrate the statement.
  5. Authors are encouraged to refer to the following papers in this review, particularly on page 3, Paucity of clinical pharmacogenetics studies in SSA: a) Sirugo G, et al. The Missing Diversity in Human Genetic Studies. Cell 2019; 177(21):26-31 https://doi.org/10.1016/j.cell.2019.02.048; and b) Asiimwe I, et al. Genetic factors influencing warfarin dose in Black-African patients: a systematic review and meta-analysis. Clin Pharmacol Ther. 2020;107(6):1420-1433. doi: 10.1002/cpt.1755.
  6. Page 12, Conclusions and future directives: Authors are asked to provide their specific recommendations to further help clinical implementation of Pgx testing in the SSA region, as they actually proposed to do on page 3, lines 73-74, Introduction, last sentence. It is unclear whether their remarks in this section are in fact suggested by the authors as solutions to properly address the current limitations across sub-Saharan African countries.
  7. Page 6, Challenges in genotyping pharmacogene variants, first paragraph, lines 166-168: Be advised that drug-metabolizing enzymes are mainly involved in elimination processes (i.e., metabolism) and, to a lesser extent, in their oral bioavailability (intestinal/hepatic first-pass effects). I would suggest rephrasing this portion as follows: “Drug response phenotypes are primarily determined by mechanisms involved in the induction or inhibition of enzymes, as well as the functionality of transporters and other proteins, involved in absorption, distribution, metabolism and…
  8. Page 6, Challenges in genotyping pharmacogene variants, first paragraph, lines 171-172: Add a reference for the following statement “Other factors affecting the activities of these enzymes include epigenetic regulation, age, lifestyle, and concomitant medications”.
  9. Please define abbreviations and acronyms the first time they are mentioned within the text (e.g., SNPs and Indels).
  10. Dihydropyrimidine dehydrogenase (DPYD) is certainly an essential gene in the pyrimidine metabolic pathway that is involved in the pharmacogenomics of fluoropyrimidine drugs (i.e., 5-fluorouracil [5-FU], capecitabine and tegafur). Patients who are carriers of these variants are considered at higher risk for severe toxicity after 5-FU administration. However, germline (e.g., DPYD*2A, c.1905+1G>A, rs3918290) and not somatic variants on this gene have been previously found to be associated with fluoropyrimidine effects (see in PharmGKB summary: fluoropyrimidine pathways. Pharmacogenet Genomics. 2011; 21(4): 237–242 at https://www.ncbi.nlm.nih.gov/pmc/articles/PMC3098754/ and the Clinical Pharmacogenetics Implementation Consortium (CPIC) Guideline for Dihydropyrimidine Dehydrogenase Genotype and Fluoropyrimidine Dosing: 2017 Update at https://cpicpgx.org/guidelines/guideline-for-fluoropyrimidines-and-dpyd/). On the other hand, the accepted abbreviation of the gene encoding for this enzyme is DPYD and not DYPD. Therefore, make corrections on page 7, first paragraph, accordingly.
  11. Page 8, last paragraph, lines 222-224: “Recently, deep NGS of the pharmacogenes of a Bantu-speaking cohort in South Africa revealed rare novel variants with predicted functional effects…” Please, provide some examples.

Reviewer 2 Report

DI have reviewed the manuscript entitled “Barriers to implementing clinical pharmacogenetics in sub-Saharan Africa” where the main goal is to address  the challenge of implementing PGx in sub-Saharan Africa with a focus on the scientific/technical barriers and to propose solutions.

The manuscript is a good review, with interesting arguments and very good information, but it needs a revision by a native speaker. There are some English mistakes on the text.

In relation to the specific points of analysis:

In the graphical abstract there are some English mistakes to solve (e.g. “Gentotyping”, .”.to implementing…”. Besides, I believe some other limitations should be included, as for example, regulatory issues, insufficient characterization of pharmacogenetic variants in Sub-Saharan Africans and others, not well addressed in the manuscript.

Some other observations are as follows:

-          The authors should include in the introduction the description of other PGx initiatives worldwide, as for example, EU-PIC (European Pharmacogenetics Implementation Consortium), the U-PGx: Ubiquitous pharmacogenomics, RELIVAF (Latin American Network for Implementation and Validation of pharmacogenomics guidelines), SEAPharma (Southeast Asian Pharmacogenomics Research Network).

-          The authors should also discuss deeplier about the influence of ethnicity in implementation of pharmacogenetic testing.

My main concern is about the lack of identification of some barriers for implementing PGx. I believe some simmilarities may have with Latin American situation (Curr Drug Metab. 2014 Feb;15(2):202-8. doi: 10.2174/1389200215666140202220753) to wide the coverage of the analysis.

Reviewer 3 Report

Emiliene B. Tata and colleagues in this review report recapped the potential limitations of implementing clinical pharmacogenetics in sub-Saharan Africa, including socio-economic perspectives as well as deficiencies in institutional structures and acceptance. However, several points should be reviewed to improve the submission, including in-depth background information inclusion and supplementation of detailed information regarding relevant genotypical data as well as the limitations within SSA from a research feasibility point of view. The unique limitations, including financial,  structure, and organization deficiencies, which are present in SSA should be further explained in other to adequately address and provide the base for future research as well as increase the value of this manuscript.

General comments:

This manuscript would benefit from English revision as there are several typos and grammatical errors that would make the content confusing in some cases, especially sentence structure and flow. Also, there are a few spelling mistakes (not including English variation in spelling by region), missing or inconsistent use of hyphens, and missing punctuation marks. Pay special attention to redundancy between sentences and paragraph structure.

Pharmacogenomics includes the use of genetic information as predictors of drug response. While the terms are treated as interchangeable at times, pharmacogenetics refers more to studies or implementations on a gene by gene basis. Consider revising the use of these terms throughout the manuscript.

Can the authors present a summarized version of regional Pgx results, including the potential mechanism of existing variants for the primary drug-metabolizing genes? It would be good to include a table using results obtained from human subjects (clinical results) and/or a separate table for results in in vitro models as this information would benefit the readers' appreciation for the uniqueness of the genotypical background of the region and how it could benefit from Pgx studies.

Abstract

Line18-21 this sentence is a little redundant and could be confusing to the reader, consider rewriting it for clarity.

Graphical abstract and other figures in the manuscript: The images used are slightly pixelated,  use higher resolution images to improve readability. 

Introduction section:

The authors could include a section explaining what the most pressing health concerns for SSA are and what are the genetic variants that could impact treatment response and success as background information to support the need for PGx testing further. As well as what type of genetic testing the authors are advocating for? Individual testing and personalized drug treatments could be unmanageable due to the region's limitations, yet large scale genotyping efforts could lead to regional genotyping and hence regional treatment design and outcome control. 

Line 46-47 authors mention over 40 Pgx tests available, due to the scope of this manuscript it would be beneficial to mention which pgx tests, as well as ADME genetic markers of interest, could be used in SSA and benefit testing implementations.

Line 52-53, the expression regarding types of interactions could be better simplified. Consider revising.

Line 56. Please include what type of clinical data could be used in conjunction with pgx testing and how this is relevant.

Line 62-65. Readers would benefit from the inclusion of information regarding these unique biomarkers and what benefits stem from their use clinically? Are they unique in resultant activities or specific to the region?

Under-resourced clinical health-care systems section:

While the authors mention the limitations of implementing Pgx testing coupled with EHR and CDSS, the reader would benefit from knowing as to what type of Pgx testing authors refer to, a brief description of their methodologies and how they have been successfully established in other African countries with similar limitations to SSA and/or suggested strategies to overcome these limitations.

Line 90: the mention of "other data" should be amended. Consider clarifying the type of data referred to instead of using this determiner.

Paucity of clinical pharmacogenetics studies in SSA section:

Line 126: previously authors had mentioned a lack of pharmacovigilance efforts in SSA as a limitation to Pgx implementations, yet they mention specific tools for pediatric pharmacovigilance, please clarify and include additional information as to what these tools are.

Line 128-131. Rare genetic variants are a challenge in any population to accurately in vivo validate due to scarce subject pools exhibiting these variants. Additionally, retrospective and prospective studies are not feasible for populations with little to no information regarding genetic variants. The studies mentioned in Table 1, though, including SSA subjects, the number of participants is limited, and these data would prove insufficient for generalized conclusions regarding the region. How do the authors suggest rare variants are validated and their drug efficacy elucidated? In vitro methodologies? Please include this additional information.

Line 150. While the authors describe how ADRs reported in the SSA appear to have unique traits when compared to other populations, including higher frequencies with certain medications, as per the scope of this study, it would be necessary to discuss why these unique traits are observed and whether they are due to specific genetic polymorphisms, environmental factors, lifestyle characteristics, etc. 

Line 164: While Figure 1 mentions the factors contributing to the under-reporting of ADRs, there is no further description of these factors thought the text. Consider omitting the figure or giving a detailed description of these factors. Descriptors such as "fear of submitting inadequate reports" seem interesting and could provide a closer look at the challenges of the region's clinicians in general; consider supplementing this information.

Challenges in genotyping pharmacogene variants section:

Line 186-188: the revision of the definition of Poor metabolizer is necessary, as the sentence structure used leads to the misunderstanding of the enzymatic metabolic capabilities of poor metabolizers. 

Line 199-208: while authors mention the importance of considering phenocoversion and cite the use of herbal medications. However, the widespread use of a known CYP2D6 inhibitor greatly challenges the interpretation of Pgx analysis and pose a risk for misclassification and the need to consider drug-induced variability as well. How do the authors suggest this be overcome? As stakeholders would question the use and applicability of population genotyping considering these pre-existing risk factors.

Line 217: To what rare variants are authors referring to? Include this information.

Line 241: arrays are not designed to detect rare regional specific variants outside the demographic in which they are designed for. Revise this statement and complement it with additional information. 

Line 257-258: complimentary genotyping assays do exist to differentiate duplicated allele hybrids, including CYP2D7. Additionally, the authors' statement is not backed or mentioned by the reference cited. Further background research is necessary for this section.

Line 268: Once more, how would the authors propose to perform in vivo analysis of rare genetic variants, when available subjects are limited?

Societal, cultural, and ethical challenges sections:

Line 299-301: stating lethargy in continuous education as the sole reason why clinicians lack confidence in recommending Pgx testing, seems a little far-fetched. If there is a study regarding this, please cite it. If not, consider mentioning the relevant underlying causes as to why Pgx testing is not widely considered. As authors have mentioned, lack of funding and availability are influential factors to impact this behavior. 

Conclusions and future directives

Line 369: As SSA rare variants with significant functional impact are of great importance in order to justify the need for Pgx testing in the region, this point should be discussed earlier in the manuscript and discuss the pertinence of the results obtained in these studies.

Line 371: How will robust large-scale studies evidence cost-effectiveness and further promote applicability?

Line 376: There are other commercially available and cost-effective techniques for the genotyping of CYP2D6 than next-generation genotyping techniques. Furthermore, this information should be included and further expanded in the introduction section.

Line 381: While CPIC and PharmKGB guidelines are an excellent source for applicability and clinical strategies, they are not the only or most applicable sources of Pgx information relevant to the region, more so considering its current limitations. Regional studies and efforts to build treatment guidelines based on in-depth studies, while scarce, should be mentioned within the manuscript as background information supporting the feasibility of performing Pgx studies in SSA.

Round 2

Reviewer 2 Report

I have reviewed the revised versión of the manuscript entitled “Barriers to implementing clinical pharmacogenetics in sub-Saharan Africa” where the main goal is to address  the challenge of implementing PGx in sub-Saharan Africa with a focus on the scientific/technical barriers and to propose solutions.

Unfortunately,  in spite of the changes made, the new version of manuscript has been poorly corrected and many mistakes has been included in the incorporated new sentences. Only, as an example, in lines 46 to 49 the sentence: “In , other expert organisations such as ous pharmacogenomics (U-46 PGx), Latin American Network for Implementation a), andutheast Asian Pharmacogenomics Researk 47 (SEAPharm) also provide pharmacogenetically-guided dosage recommendations as reviewed 48 lsewhere ADDINError! Bookmark not defined.|[6]}.”  This sentence has many mistakes. I don’t know what U-46 PGx is. I believe this sentence should be like this :“Other expert organisations such as EU-PIC (European Pharmacogenetics Implementation Consortium), the U-PGx: Ubiquitous pharmacogenomics, RELIVAF (Latin American Network for Implementation and Validation of pharmacogenomics guidelines) and SEAPharma (Southeast Asian Pharmacogenomics Research Network), are also directed to provide pharmacogenetically-guided dosage recommendations as reviewed elsewhere [6]”.However, the reference 6 has nothing to do with this topic.

As there are many mistakes, it has no sense to me, to include a detailed description of these typographic errors. The authors should review the whole manuscript in a professional way.

Besides, some of my observations were not addressed, for example, the discussion about  the influence of ethnicity in implementation of pharmacogenetic testing and the identification of some other potential barriers. A comparison with Latin American and other regions situation should be desirable (doi: 10.2174/1389200215666140202220753).

Reviewer 3 Report

The revision has been done very well, and I have no further points.

Author Response

No comments from reviewer

Round 3

Reviewer 2 Report

Dear Editor

I feel that the manuscript now is in appropriate form to be published. At least, my  concerns have been overcome.

Luis A. Quiñones